# Early Blood Glucose Level Post-Admission Correlates with the Outcomes and Oxidative Stress in Neonatal Hypoxic-Ischemic Encephalopathy

**DOI:** 10.3390/antiox11010039

**Published:** 2021-12-24

**Authors:** Inn-Chi Lee, Jiann-Jou Yang, Ying-Ming Liou

**Affiliations:** 1Division of Pediatric Neurology, Department of Pediatrics, Chung Shan Medical University Hospital, Taichung 40201, Taiwan; 2Institute of Medicine, School of Medicine, Chung Shan Medical University, Taichung 40201, Taiwan; jiannjou@csmu.edu.tw; 3Genetics Laboratory and Department of Biomedical Sciences, Chung Shan Medical University, Taichung 40201, Taiwan; 4Department of Life Sciences, National Chung-Hsing University, Taichung 40227, Taiwan; ymlion@dragon.nchu.edu.tw; 5The iEGG and Animal Biotechnology Center and Rong Hsing Research Center for Translational Medicine, National Chung Hsing University, Taichung 40227, Taiwan

**Keywords:** newborns, hypoxic-ischemic encephalopathy, biomarker, thalamus, basal ganglion, glucose, hearing, MRI, outcomes, oxidative stress

## Abstract

The antioxidant defense system is involved in the pathogenesis of neonatal hypoxic-ischemic encephalopathy (HIE). To analyze the relationship between first serum blood glucose levels and outcomes in neonatal HIE, seventy-four patients were divided, based on the first glucose level, into group 1 (>0 mg/dL and <60 mg/dL, *n* =11), group 2 (≥60 mg/dL and <150 mg/dL, *n* = 49), and group 3 (≥150 mg/dL, *n* = 14). Abnormal glucose levels had poor outcomes among three groups in terms of the clinical stage *(p =* 0.001), brain parenchymal lesion *(p* = 0.004), and neurodevelopmental outcomes *(p* = 0.029). Hearing impairment was more common in group 3 than in group 1 (*p* = 0.062) and group 2 (*p* = 0.010). The MRI findings of group 3 exhibited more thalamus and basal ganglion lesions than those of group 1 *(p* = 0.012). The glucose level was significantly correlated with clinical staging (*p*
*<* 0.001), parenchymal brain lesions (*p* = 0.044), hearing impairment (*p* = 0.003), and neurodevelopmental outcomes (*p* = 0.005) by Pearson’s test. The first blood glucose level in neonatal HIE is an important biomarker for clinical staging, MRI findings, as well as hearing and neurodevelopment outcomes. Hyperglycemic patients had a higher odds ratio for thalamus, basal ganglia, and brain stem lesions than hypoglycemic patients with white matter and focal ischemic injury. Hyperglycemia can be due to prolonged or intermittent hypoxia and can be associated with poor outcomes.

## 1. Introduction

Birth asphyxia is a physiological derangement seen in newborn infants due to a prolonged or profound mismatch between oxygen demand and oxygen delivery [1,2,3,4]. It can cause mild to severe neurodevelopmental disabilities. Moderate to severe asphyxia can cause irreversible cerebral cell damage, neonatal seizure, and death, leading to a syndrome of hypoxic-ischemic encephalopathy (HIE) that has multi-organ involvement. Rescue hypothermia has been proven effective and has few adverse effects on newborns with HIE [5,6,7], and used to reduce neurological injury; nevertheless, a 45–55% risk of death or moderate–severe disability remains in treated infants [5,6,8]. Rescued hypothermia therapy has brought pressure on clinicians to make an early and accurate assessment of neonatal HIE and predict the severity of encephalopathy that will ensue [9]. Although hypothermia therapy is proven effective in moderate and severe neonatal HIE [5,10,11], it has not been proven to be beneficial for mild HIE [12,13,14]. Adjunctive tools or biomarkers for the optimal assessment of neonatal HIE are needed for early diagnosis and timely treatment.

There are two distinctive mechanisms involved in HIE. The first is hypoxic injury. The brain is susceptible to hypoxia, particularly in regions, such as the hippocampus, basal ganglion, thalamus, and brain stem [15,16]. This is called selective neuronal necrosis and status marmoratus of the basal ganglia and thalamus [17,18]. The second is ischemic change. This change occurs due to ischemic injury caused by hypotension or focal infarct, including the anterior, middle and posterior cerebral arteries, as well as their branches [19,20]. The pathogenic mechanisms underlying neonatal HIE can be categorized into three phases. The first phase involves primary energy failure due to the hypoxic-ischemic injury, the secondary phase is a consequence of reoxygenation and reperfusion, and the third phase wherein the hypoxic-ischemic injury can worsen and the inflammation can turn into a subacute and chronic condition [21,22,23,24,25]. The antioxidant defense system is involved in the pathogenesis of neonatal HIE, particularly in the aforementioned second and third phases [26,27,28]. During the second phase, the activity of the antioxidant defense system is exhausted due to oxidative stress, leading to further damage, including lipid peroxidation, protein denaturation, enzyme inactivation, and DNA damage [29,30,31]. Glucose concentration can affect the oxidant–antioxidant balance system in the second and third phases, and impair the antioxidant defense system. Thus, glucose imbalance, including hyperglycemia or hypoglycemia, is presumed to play an important role in neonatal HIE and is also a potential diagnostic and prognostic biomarker.

The normal glucose concentration in the blood of newborn infants is between 2.5 mmol/L (45 mg/dL) and 7.0 mmol/L (126 mg/dL). Most newborns have a blood glucose concentration in the middle of the normal range, approximately 3.5 mmol/L (63 mg/dL) to 5 mmol/L (90 mg/dL) [32]. Hypoxia can cause enhanced or unchanged glucose levels and even decreased or increased concentrations of blood glucose, serum insulin, and plasma glucagon [33,34,35,36,37]. These changes occurred since hypoxia was found to stimulate insulin secretion from newborn rats but was inhibited in juvenile rats [38]. The fasting blood glucose concentration remained unchanged in response to acute hypoxia (hours) [33], but increased after 3 days of hypoxia [39,40]. 

A retrospective cohort study of neonates with encephalopathy showed that hypoglycemia was more often associated with watershed hypoxic-ischemic brain injury than basal ganglia hypoxic-ischemic injury [41], which has been supported in several studies [42,43]. In a prospective cohort analysis of neonates with HIE treated with hypothermia, hypoglycemia in neonatal HIE was associated with an increased odds ratio in a watershed or focal–multifocal brain injury [42]. Hypoglycemia has effects on brain injury, resulting from a hypoxia-ischemia mechanism and is associated with worsening corticospinal tract injury, leading to declined motor and cognitive outcomes at 1 year of age [43]. Regarding hyperglycemia, several retrospective studies [44,45] have reported that hyperglycemia and glucose variability are associated with an increased risk of death and major disability. In hypothermia-treated neonates with HIE, early hypoglycemia or hyperglycemia was noted to cause hearing impairment [46]. In a study of 214 treated neonates with HIE, hyperglycemia was associated with death or unfavorable outcomes at 18 months [44]. Hypoglycemia and hyperglycemia after birth are significant factors that correlate with complications; however, there are also notable differences between the outcomes of glucose abnormalities.

Early diagnosis and rapid treatment are critical for the long-term prognosis of neonatal encephalopathies. Identifying the glucose level to correlate with clinical staging and outcomes can help clinicians begin early treatment. In this study, the first glucose level after the first admission was obtained to correlate with clinical staging, hearing outcomes, magnetic resonance imaging (MRI) findings, and neurodevelopmental outcomes for early diagnosis of neonatal HIE.

## 2. Patients and Methods

### 2.1. Patients

We retrospectively reviewed the patient charts of neonates diagnosed with HIE based on a clinical history of fetal distress, metabolic acidosis, or positive-pressure ventilation immediately after birth, at Chung Shan Medical University Hospital from 2015 to 2020. The clinical stages of HIE were classified as Sarnat stage I (mild), II (moderate), and III (severe) [5,6]. Blood glucose levels were measured at the time of admission.

Further examinations for HIE, including head *ultrasound* (HUS), MRI, automated electrocardiography (aEEG), continuous neonatal conventional EEG monitoring, hearing testing (automated *auditory* brainstem response (aABR)), and *auditory* brainstem response (ABR) testing, were performed before discharge. For stage I patients, HUS was performed at birth and at 1, 3, 7, and 14 days of age. An MRI was performed if the clinical condition was suspected to be a brain lesion among these patients. 

An experienced pediatric neurologist and neonatologist consultant divided the patients into group 1, classified as Sernat stage I (mild) HIE, and group 2, classified as stage II (moderate) and III (severe). The differences in blood biomarker levels were compared between the two groups (Figure 1).

The HIE patients were then divided into three groups according to the first glucose level after the first admission after birth: group 1 (>0 mg/dL and <60 mg/dL, *n* = 11), group 2 (≥60 mg/dL and <150 mg/dL, *n* = 49), and group 3 (≥150 mg/dL, *n* = 14). The analysis of the three-group outcomes were based on short-term (clinical staging, hearing test, and MRI findings) and long-term neurodevelopmental changes at 1 year.

### 2.2. MRI Classification

The MRI findings were divided into two groups to study the correlation between the biomarkers and MRI changes. The first group showed no brain lesions in the parenchyma, while the second group showed brain lesions in the parenchyma. In the second group, brain MRI was classified into two subgroups based on the location of the lesions: one of the basal ganglia, thalamus, or brain stem (midbrain, pons, and lower brain stem), and the other involved areas other than the basal ganglion, thalamus, and brain stem.

### 2.3. Hearing Tests before First Discharge 

For patients who failed the aABR test twice during universal newborn screening, ABR testing, otoacoustic emissions, and steady-state evoked potentials were performed [47]. The ABR waveforms were analyzed, and the latency of peak V was defined and adjusted by an experienced pediatric neurologist or otolaryngologist. The degree of hearing loss was classified as normal (<25 and  ≤35 dB nHL) and abnormal, including mild (>35 and  ≤45 dB nHL), moderate (>45 and  ≤65 dB nHL), severe (>65 and  ≤90 dB nHL), or profound (>90 dB nHL) [48,49].

### 2.4. Measurement of Neurodevelopmental Outcome at >1 Year of Age

The third edition of the Bayley Scales of Infant and Toddler Development (Bayley-III) was used to evaluate the neurodevelopmental outcomes at >1 year of age. Cognitive and motor subscales were used to interpret the neurodevelopmental outcomes. The Bayley-III scores were defined as follows: normal if both cognitive and motor subscale scores were ≥85, and abnormal if one of the cognitive and motor subscale scores was <85 [17,18].

### 2.5. Statistical Analysis

The independent t-test was performed to compare the means of two independent groups for the significant differences between groups, and the categorical variables were analyzed using the chi-square test. The Fisher’s exact test was performed when the sample size was small. The odds ratio (OR) was calculated by dividing the odds of the first group by the odds of the second group. Furthermore, the Mann–Whitney U test was performed if the sample distribution was nonparametric, and the statistical significance was set at a *p*-value of < 0.05. For correlation analyses, Pearson’s test was performed to measure the strength of the linear association between the two variables. All statistical tests were performed using SPSS (version 14.0; SPSS Institute, Chicago, IL, USA).

Ethical approval for the study was provided by Chung Shan Medical University Hospital’s internal review board (IRB #: CS14003) and was performed in accordance with the relevant guidelines.

### 2.6. Informed Consent Statement

Since this is a retrospective study, informed consent was not required.

## 3. Results

### 3.1. Demographic Data in Newborns with HIEs

After excluding 18 patients due to congenital anomalies (*n* = 7), preterm with a gestational age less than 36 weeks (*n* = 10), or with a confirmed genetic defect later on (*n* = 1), 74 patients with HIE were enrolled. Eleven belonged to group 1, 49 belonged to group 2, and 14 belonged to group 3 (Figure 1). Among the three groups, factors, including birth weight, sex, age, and inborn or outborn method of delivery (cesarean section or vaginal delivery), were not significant (Table 1). The 1 min and 5 min Apgar scores were not significantly different in the three groups (Table 1). The initial blood glucose level is shown in Figure 2.

### 3.2. Clinical Staging and Glucose Level

Among the patients in group 1, 2 (18.2%) had stage I, 7 (63.6%) had stage II, and 2 (18.2%) had stage III HIE. In 49 cases in group 2, 27 (55.1%) had stage I, 14 (28.6%) had stage II, and 8 (16.3%) had stage III HIE disease. Among the 14 cases in group 3, 1 (7.1%) had stage I, 6 (42.9%) had stage II, and 7 (50.0%) had stage III HIE (Figure 1 and Table 1). The differences were significant among the group with clinical staging (χ2 (4, *n* = 74) = 16.5, *p* = 0.002). However, in groups 1 and 3, the differences in clinical staging distribution were not significant (χ2 (2, *n* = 25) = 2.9, *p* = 0.238) (Table 2). Glucose levels were significantly correlated with clinical staging (r (72) = 0.379, *p* < 0.001).

### 3.3. Correlation of Parenchymal Brain Lesion and Glucose Level

Lesions in brain parenchyma by MRI, computed tomography, or ultrasound were detected in 7 (63.6%) patients in group 1, 12 (24.5%) patients in group 2, and 9 (64.3%) in group 3 (Figure 3). The differences in the brain parenchymal lesions between the 3 groups were significant (χ2 (2, *n* = 74) = 11.0, *p* = 0.004). When groups 1 and 3 were compared, the brain parenchymal lesions were not significantly different (χ2 (1, *n* = 25) = 0.001, *p* = 0.97) (Table 2). The glucose levels were significantly correlated with the parenchymal brain lesions (r(72) = 0.238, *p* = 0. 044)).

The imaging study in group 1 showed that 1 (9.1%) out of 11 cases exhibited thalamus or basal ganglion lesions. Regarding the remaining 10 cases in group 1, 4 (36.4%) had normal brain findings and 6 (54.5%) exhibited white matter lesions or focal brain lesions that did not involve the thalamus and basal ganglia. The findings were compared with those of group 3 patients, in whom 9 (64.3 %) out of 14 had a thalamus, basal ganglion, or brain stem lesion. The difference in involving the thalamus, basal ganglion, and brain stem between groups 1 and 3 was significant (χ2 (1, *n* = 16) = 8.9, *p* = 0.002) (Table 2). Of the 28 patients with abnormal lesions on the brain MRI scans, the glucose level was significantly correlated with the locations of the brain lesions on MRI scans (r(26) = 0.698, *p* < 0.001).

### 3.4. Correlation of Hearing Impairments and Glucose Level

Hearing impairment with neonatal HIE were 1 (8.1%) in group 1, 6 (12.2%) in group 2, and 6 (42.9%) in group 3. The hearing impairment across the three groups were significantly different (χ2 (2, *n* = 74) = 7.7, *p* = 0.021). Hearing impairment was more common in group 3 than in group 1 (χ2 (1, *n* = 25) = 3.5, *p* = 0.062) and group 2 (χ2 (1, *n* = 63) = 6.6, *p* = 0.010). Of the 72 patients with HIE, glucose level was significantly correlated with hearing impairment (r(72) = 0.341, *p* = 0.003).

### 3.5. Correlation of Neurodevelopmental Outcomes and Glucose Level

The neurodevelopmental outcome at at least 1 year of age was correlated with the first glucose level after admission. In group 1, 5 (45.5%) had unremarkable and 6 (54.5%) had abnormal neurodevelopmental outcomes. In group 2, 35 (71.4%) had unremarkable and 14 (28.6%) had abnormal neurodevelopmental outcomes. In group 3, 5 (35.7%) had unremarkable and 9 (64.3%) had abnormal neurodevelopmental outcomes. The neurodevelopmental outcomes in the three groups were significantly different (χ2 (2, *n* = 74) = 7.11, *p* = 0.029). However, the ratio of abnormal neurodevelopmental outcomes at >1 year of age in groups 1 and 3 was not significant (χ2 (1, *n* = 25) = 0.24, *p* = 0.622) (Table 2). In the 72 patients with HIE, the glucose levels were significantly correlated with the neurodevelopmental outcomes (r(72) = 0.331, *p* = 0.005).

### 3.6. The Differences of Other Blood Biomarkers in the Group 1, Group 2, and Group 3 Patients

We compared groups 1 and 3 and found significant difference in the levels of lactate dehydrogenase (LDH) (3560.4 ± 2851.1 vs. 863.3 ± 510.3; U = 60; *p* = 0.001); serum aspartate transaminase (SGOT) (504.0 ± 539.0 vs. 113.0 ± 89.9; U = 90; *p* = 0.019); serum alanine transaminase (SGPT) (162.8 ± 163.2 vs. 33.4 ± 30.3; U = 84, *p* = 0.009); platelets (164,888.9 ± 58,650.3 vs. 246,714.3 ± 67,271.5 mm^3^ µL; U = 87; *p* = 0.007); C-reactive protein (4.3 ± 5.9 vs. 0.04 ± 0.2 mg/L; U = 122; *p* = 0.025); and creatine kinase (CK) (5080.1 ± 7238.7 vs. 1830.2 ± 2808.8 U/L; U = 106; *p* = 0.047) (Table 3). The glucose levels were significantly correlated with LDH (r(64) = −0.401, *p* < 0.001); SGPT (r(62) = −0.354, *p* = 0.005), SGPT (r(62) = −0.324, *p* = 0.010); platelet count (r(67) = 0.208, *p* = 0.086); and CK (r(67) = −0.235, *p* = 0.066). However, the lactate levels; white blood cell counts; hemoglobin levels, blood urea nitrogen levels; creatinine levels; prothrombin time; activated partial thromboplastin time; and albumin, sodium, potassium, creatine kinase-MB, and troponin levels were not significant (Table 3). The findings highlight that the systemic biomarker levels were higher in group 1 than in group 3.

## 4. Discussion

A significant contribution of this study is the correlation of the first glucose level of neonates with HIE with clinical staging, findings of brain MRI, hearing outcomes, and neurodevelopmental outcomes at 1 year. Hypoglycemia and hyperglycemia are associated with advanced staging, brain parenchymal lesions, hearing impairment, and abnormal neurodevelopmental outcomes. Hyperglycemia was strongly related to hearing loss and thalamus, basal ganglia, and brain stem lesions; however, hypoglycemia was closely related to white matter lesions. This finding is also compatible with high systemic biomarker levels indicating liver injury (LDH, SGPT, SGPT, and platelets) in the hypoglycemic group. This finding supports the hypothesis that hyperglycemia is caused by prolonged or intermittent hypoxia. In addition, cases with increased blood glucose levels can be more severe than those with hypoglycemia due to the involvement of the thalamus, basal ganglia, and brain stem, especially the 8th nuclei with hearing impairments.

As hypothermia therapy needs to be performed in a timely manner, the use of a simple and convenient method, such as blood glucose measurement, can be useful in the early prediction of the staging of neonatal HIE and the need for the initiation of treatment. Although a combination of other biomarkers, such as lactate and LDH levels, can help predict the severity of HIE, obtaining the glucose level is a rapid and convenient method. This is beneficial for management, as it allows early rescue hypothermia performed 6 h after birth and permits the use of neuroprotective drugs.

Despite similar clinical staging and MRI findings, hearing impairment in patients with hyperglycemia with a first blood glucose level >150 mg/dL is worse than that in patients with hypoglycemia, with a first blood glucose level <60 mg/dL. This finding can be explained by several hypotheses. First, hypoglycemia can further induce fatty acid oxidation and cause ketosis, which can have a protective effect in the brain; hyperglycemia does not have this effect. Second, hypoxia demonstrated significant increases in plasma glucose and insulin [38]; however, intermittent or prolonged hypoxia can increase insulin resistance in genetically obese mice [35] that causes reflex hyperglycemia. Therefore, further studies on the effects of glucose abnormalities to neonatal HIE outcomes are warranted. Third, hypoglycemia can be induced by hyperinsulinemia due to neonatal HIE; however, the compensation mechanism of hyperglycemia in HIE can be exhausted, reflecting the poor condition seen in newborns.

In neonates with encephalopathy, periods of hyperglycemia were common and temporally associated with worse aEEG background scores, reduced sleep–wake cycling, and increased electrographic seizures, including after adjusting for clinical markers of hypoxia-ischemia. Hyperglycemia epochs were also associated with poor aEEG background scores, including after adjusting for hypoxia-ischemia severity. Our data support the hypothesis that the proactive avoidance of hyperglycemia can be a neuroprotective strategy for infants with neonatal encephalopathy [50]. Hypoglycemic or hyperglycemic blood levels can affect MRI findings. In hypoglycemia, the watershed or focal multifocal infarcts were observed on MRI scans [43], whereas hyperglycemia was more associated with the basal ganglia or global injury findings on MRI scans [43]. In 56 neonatal HIE, all of whom died, we studied their neurodevelopmental outcomes and first 24 h glucose level and highlighted 9 patients with first glucose levels over 200 mg/dL.

The findings of the aforementioned studies [28,29] were compatible with our findings that hypoglycemia and hyperglycemia can increase the risk of poor outcomes in neonatal HIE based on MRI findings. However, in our study, we highlighted that hyperglycemia was associated with a high risk of hearing impairment, which is crucial for childhood neurodevelopment. In hypothermia-treated neonates with HIE for 42 babies, 4 (9.5%) had hearing impairments. The development of hearing loss was associated with abnormal blood glucose levels, low Apgar scores, and evidence of multi-organ dysfunction and increased SGPT and SGPT levels [46], which are compatible with our findings. In addition, we also highlighted that the hyperglycemic patients had more thalamic and basal ganglion injuries than those with hypoglycemia before the first 6 h. These findings suggest that hyperglycemia can cause selective neuronal necrosis that causes injury to susceptible brain tissue, including the basal ganglia, thalamus, and brain stem. We hypothesized that the mechanism of neonatal HIE is related to glucose and clinical staging (Figure 4). Hyperglycemia in the reoxygenation and reperfusion stage can lead to further brain injury due to the consequence of oxidation stress. Hypoglycemia can cause ketogenesis by acting as an alternative cerebral fuel and as antioxidants. This can explain why the hypoglycemia group had better outcomes than the hyperglycemia group in the study. Hyperglycemia caused by insulin resistance can contribute to further brain injury as the consequence of oxidation stress that can be a useful biomarker of poor neurological outcomes and worse neurological consequences [51]. Thus, avoiding hyperglycemia after admission is mandatory in the clinical management of neonatal HIE.

However, this study has some limitations. We presented a limited number of HIE cases. Our findings can be biased and comprised owing to the fewer cases than needed for reliable results. Therefore, further studies with an increased number of cases are warranted. Furthermore, in the HIE stage I group with favorable outcomes, an aggressive image study is not available from the national insurance agency in Taiwan. However, a series of HUS can support the imaging findings, and a clinical follow-up of up to 1 year can diagnose the patients without significant brain parenchymal lesions.

## 5. Conclusions

The first blood glucose level is an important biomarker for clinical staging, MRI findings, hearing impairment, and neurodevelopmental outcomes in neonatal HIE. Hyperglycemic patients had higher ORs in the thalamus, basal ganglia, and brain stem lesions than hypoglycemic patients who were often related to white matter and focal ischemic injury. This finding supports the fact that hyperglycemia possibly occurred due to prolonged or intermittent hypoxia and oxidation stress, and led to worse outcomes due to the involvement of the thalamus and basal ganglia. In neonatal HIE, early glucose levels after the first admission can be a rapid and convenient biomarker for a timely diagnosis and early treatment administration.

## Figures and Tables

**Figure 1 antioxidants-11-00039-f001:**
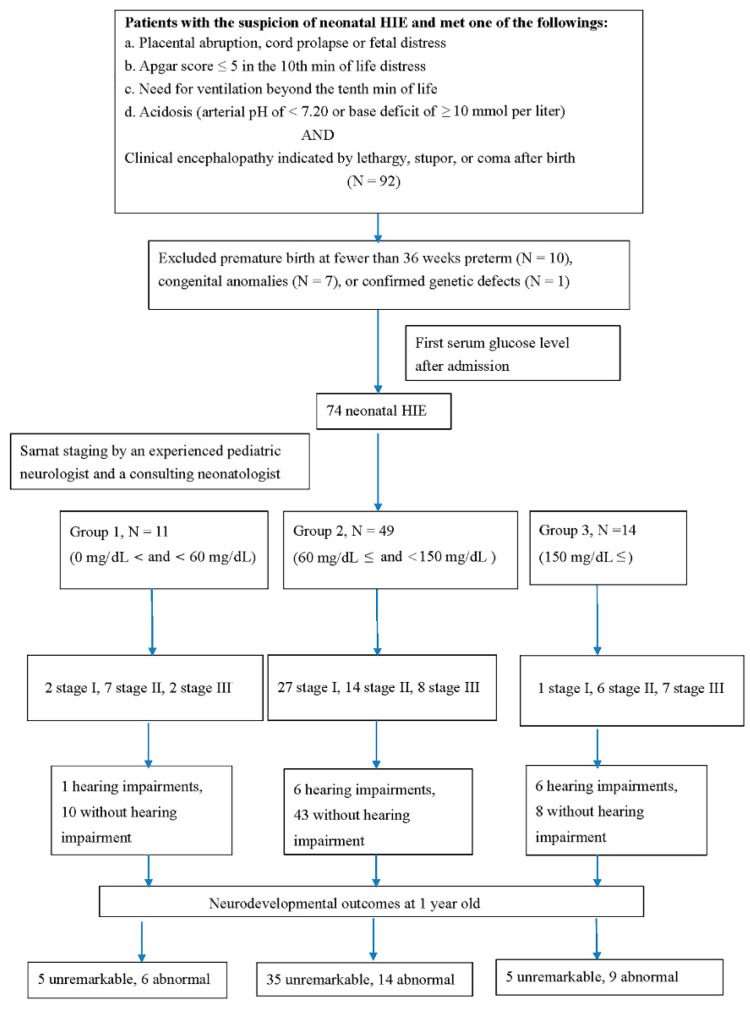
Flow chart of the study procedure demonstrated in neonatal hypoxic-ischemic encephalopathy cases and their first glucose level after admission. MRI, magnetic resonance imaging.

**Figure 2 antioxidants-11-00039-f002:**
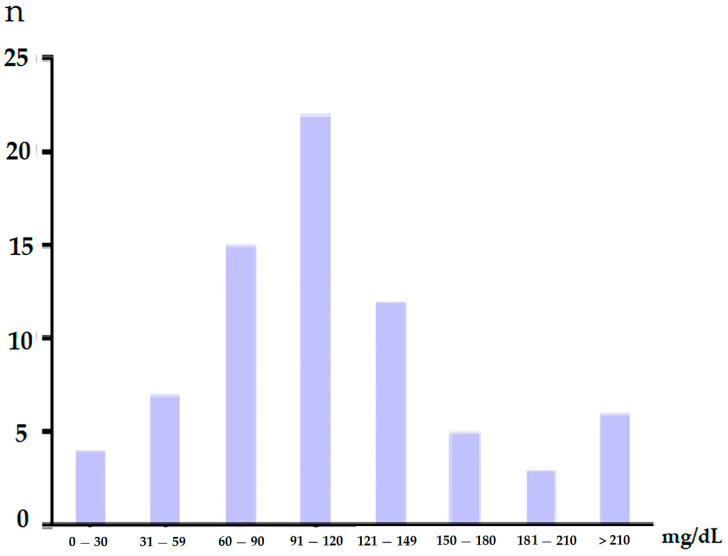
The initial blood glucose level.

**Figure 3 antioxidants-11-00039-f003:**
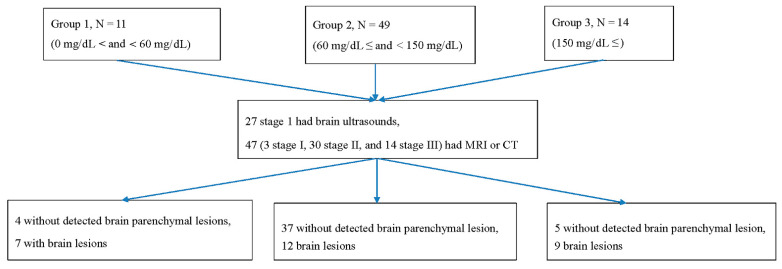
The correlation of the glucose level in the three groups with imaging findings.

**Figure 4 antioxidants-11-00039-f004:**
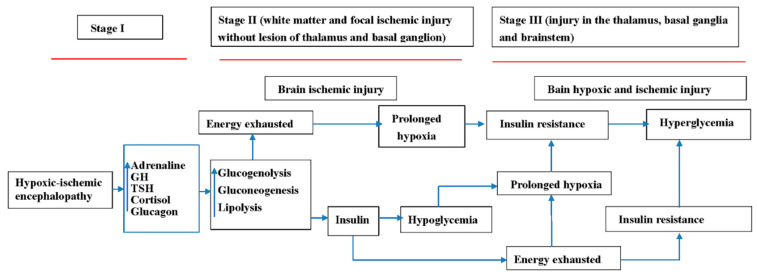
The hypothesized mechanism of hypoglycemia and hyperglycemia in neonatal hypoxic-ischemic encephalopathy.

**Table 1 antioxidants-11-00039-t001:** Seventy-four neonatal hypoxic-ischemic encephalopathy cases were classified into three groups, according to the first serum glucose level taken before 6 h of birth.

First Glucose Level after Admission	>0 mg/dL and <60 mg/dL	≥60 mg/dL and <150 mg/dL	≥150 mg/dL	*p* between Group 1 and Group 2	*p* between Group 2 and Group 3	*p* between Group 1 and Group 3
	(Group 1, n = 11)	(Group 2, n = 49)	(Group 3, n = 14)			
Mean ± SD (Range)	35.1 ± 19.3 (9.0–58.0)	104.4 ± 23.4 (60.0–145.0)	222.0 ± 62.4 (152.0–332.0)			
Gestational age (weeks)	38.3 ± 2.0	38.7 ± 1.3	38.6 ± 1.2	t(58) = −1.13, *p* = 0.263	t(61) = −0.628, *p* = 0.533	t(23) = −0.47, *p* = 0.642
Birth weight (gm)	3251.2 ± 729.1	2972.2 ± 368.6	2798.3 ± 446.2	t(58) = 1.77, *p* = 0.082	t(61) = −1.4, *p* = 0.164	t(23) = 1.82, *p* = 0.084
Gender						
Male	6 (54.5%)	27 (55.1%)	8 (57.1%)	χ2 (1, *n* = 60) = 0.01, *p* = 0.973	χ2 (1, *n* = 63) = 0.02, *p* = 0.892	χ2 (1, *n* = 25) = 0.02, *p* = 0.897
Female	5 (45.5%)	22 (44.9%)	6 (42.9%)			
Inborn	5 (45.5%)	19 (38.8%)	5 (35.7%)	χ2 (1, *n* = 60) = 0.17, *p* = 0.683	χ2 (1, *n* = 63) = 0.04, *p* = 0.835	χ2 (1, *n* = 25) = 0.24, *p* = 0.623
Outborn	6 (54.5%)	30 (61.2%)	9 (64.3%)			
Method of delivery						
Cesarean section	4 (36.4%)	18 (36.7%)	5 (35.7%)	χ2 (1, *n* = 60) = 0.01, *p* = 0.982	χ2 (1, *n* = 63) = 0.01, *p* = 0.944	χ2 (1, *n* = 25) = 0.01, *p* = 0.973
Vaginal delivery	7 (63.6%)	31 (63.3%)	9 (64.3%)			
Apgar score at one minute	4.4 ± 2.1	3.7 ± 2.1	3.5 ± 2.9	*t*(58) = 0.74, *p* = 0.466	*t*(61) = 0.45, *p* = 0.656	*t*(23) = 1.01, *p* = 0.325
Apgar score at five minutes	6.3 ± 2.1	5.4 ± 2.4	4.7 ± 2.6	*t*(58) = −0.86, *p* = 0.397	*t*(61) = 0.93, *p* = 0.358	*t*(23) = 1.85, *p* = 0.078

HIE, hypoxic-ischemic encephalopathy and SD, standard deviation.

**Table 2 antioxidants-11-00039-t002:** The clinical staging, hearing outcomes, imaging findings, and neurodevelopmental outcomes in the three groups with neonatal hypoxic-ischemic encephalopathy.

	Group 1, *n* = 11	Group 2, *n* = 49	Group 3, *n* = 14	*p* Values Among Group 1, Group 2, and Group 3	*p* Values between Group 1 and Group 3
Clinical staging					
Stage I (*n* = 30)	2 (18.2%)	27 (55.1%)	1 (7.1%)	**χ2 (4, *n* = 74) = 16.5,** ** *p* ** **= 0.002**	χ2 (2, *n* = 25) = 2.9, *p* = 0.238
Stage II (*n* = 22)	7 (63.6%)	14 (28.6%)	6 (42.9%)		
Stage III (*n* = 22)	2 (18.2%)	8 (16.3%)	7 (50.0%)		
Image finding ^+^					
Patients without detected lesion in brain parenchyma (*n* = 46)	4 (36.4%)	37 (75.5%)	5 (35.7%)	**χ2 (2, *n* = 74) = 11.0,** ***p* = 0.004**	χ2 (1, *n* = 25) = 0.001, *p* = 0.97
Patients with detected lesion in brain parenchyma (*n* = 28)	7 (63.6%)	12 (24.5%)	9 (64.3%)		
Abnormal MRI or CT					
Basal ganglion, thalamus, and brain stem (*n* = 20)	1 (14.3%)	10 (83.3%) (8 cases > 110 mg/ dL)	9 (100%)	**χ2 (2, *n* = 28) = 12.3,** ***p* = 0.002**	**χ2 (1, *n* =** **16) = 8.9,** ***p* = 0.002**
White matter or focal ischemic injury without lesion of basal ganglion, thalamus, and brain stem (*n* = 8)	6 (85.7%)	2 (16.7%)	0 (0%)		
Hearing outcomes					
Patients with hearing impairments (*n* = 13)	1 (8.1%)	6 (12.2%)	6 (42.9%)	**χ2 (2, *n* = 74) = 7.7,** ***p* = 0.021**	χ2 (1, *n* = 25) = 3.5, *p* = 0.062
Patients without hearing impairments (*n* = 61)	10 (91.9%)	43 (87.8%)	8 (57.1%)		
Neurodevelopmental outcomes at 1 year old					
Unremarkable (*n* = 45)	5 (45.5%)	35 (71.4%)	5 (35.7%)	**χ2 (2, *n* = 74) = ** **7.11,** ** *p* ** **= 0.029**	χ2 (1, *n* = 25) = 0.24, *p* = 0.622
Abnormal (*n* = 29)	6 (54.5%)	14 (28.6%)	9 (64.3%)		

MRI, magnetic resonance imaging; CT, computed tomography. ^+^ The findings of the image included 27 stage 1 with brain ultrasounds, and 47 (3 stage I, 30 stage II, and 14 stage III) with brain MRI or CT. Bold fonts indicate significance.

**Table 3 antioxidants-11-00039-t003:** Biomarkers exhibited in group 1, group 2, and group 3.

Biomarkers	Group 1, *n* = 11	Group 2, *n* = 49	Group 3, *n* = 14	*p* Values between Group 1 and Group 2	*p* Values between Group 2 and Group 3	*p* Values between Group 1 and Group 3
WBCs (9100–34,000 mm^3^ µL)	23,372.0 ± 17,485.5	19,784.2 ± 7501.3	23,688.6 ± 8536.4	U = 199, *p* = 0.935	U = 214, *p* = 0.072	U = 41, *p* = 0.166
Platelet (84–478 mm^3^ µL)	164,888.9 ± 58,650.3	238,977.8 ± 74,946.5	246,714.3 ± 67,271.5	**U = 87, *p* = 0.007**	U = 297, *p* = 0.748	**U = 21, *p* = 0.008**
Hemoglobin (13.88 ± 1.34 g/dL)	17.3 ± 2.6	16.8 ± 2.1	18.9 ± 12.1	U = 160, *p* = 0.318	U = 256, *p* = 0.297	U = 42, *p* = 0.176
SGOT (30–100 U/L)	504.0 ± 539.0	129.8 ± 160.4	113.0 ± 89.9	**U = 90, *p* = 0.019**	U = 276, *p* = 0.945	**U = 31, *p* = 0.044**
SGPT (6–40 U/L)	162.8 ± 163.2	36.5 ± 58.9	33.4 ± 30.3	**U = 84, *p* = 0.009**	U = 244, *p* = 0.411	**U = 32, *p* = 0.046**
BUN (3–12 mg/dL)	11.2 ± 4.1	10.7 ± 3.8	11.9 ± 5.7	U = 173, *p* = 0.691	U = 264, *p* = 0.568	U = 63, *p* = 0.975
Creatinine (0.03–0.50 mg/dL)	1.0 ± 0.2	0.9 ± 0.2	1.1 ± 0.3	U = 156, *p* = 0.472	U = 198, *p* = 0.085	U = 54, *p* = 0.549
Lactate (4.4 to 14.4 mg/dL)	86.9 ± 76.8	68.6 ± 46.9	92.5 ± 33.7	U = 213, *p* = 0.964	**U = 173, *p* = 0.017**	U = 55, *p* = 0.403
LDH (170–580 U/L)	3560.4 ± 2851.1	875.7 ± 657.8	863.3 ± 510.3	**U = 60, *p* = 0.001**	U = 300, *p* = 0.985	**U = 22, *p* = 0.009**
PT (13.0 ± 1.43 s)	19.8 ± 15.8	16.4 ± 6.1	16.8 ± 3.0	U = 179, *p* = 0.726	U = 235, *p* = 0.217	U = 60, *p* = 0.850
aPTT (42.9 ± 5.80 s)	62.4 ± 23.7	56.0 ± 25.5	71.0 ± 23.0	U =155, *p* = 0.352	**U = 173, *p* = 0.018**	U = 46, *p* = 0.284
Albumin (2.5–3.4 g/dL)	3.3 ± 0.8	3.5 ± 0.4	3.7 ± 0.3	U = 164, *p* = 0.363	U = 236, *p* = 0.294	U = 44, *p* = 0.315
Na (133–146 mmol/L)	135.6 ± 1.8	136.0 ± 3.6	135.1 ± 3.9	U = 182, *p* = 0.631	U = 230, *p* = 0.127	U = 46, *p* = 0.262
K (3.2–5.5 mmol/L)	3.8 ± 0.6	4.1 ± 0.7	4.2 ± 0.7	U = 155, *p* = 0.269	U = 294, *p* = 0.714	U = 46, *p* = 0.218
CK (39–308 U/L)	5080.1 ± 7238.7	1986.1 ± 2398.8	1830.2 ± 2808.8	**U = 106, *p* = 0** **.047**	U = 225, *p* = 0.401	**U = 29, *p* = 0.043**
CK-MB (0–4.5 ng/mL)	75.9 ± 70.8	68.7 ± 94.5	23.7 ± 19.7	U = 61, *p* = 0.736	U = 64, *p* = 0.084	U = 11, *p* = 0.188
Troponin I (0–30 pg/mL)	915.7 ± 1365.1	154.9 ± 420.0	97.8 ± 62.3	U = 77, *p* = 0.076	U = 195, *p* = 0.459	U = 28, *p* = 0.236
CRP (1.5–20 mg/L)	4.3 ± 5.9	5.1 ± 19.6	0.04 ± 0.2	**U = 122, *p* = 0** **.025**	U = 249, *p* = 0.102	**U = 23, *p* = 0.002**

Bold fonts indicate *p* < 0.05. HIE, hypoxic-ischemic encephalopathy; ST, standard deviation; WBCs, white blood cells; GOT, aspartate transaminase; GPT, alanine transaminase; BUN, blood urea nitrogen; LDH, lactate dehydrogenase; PT, prothrombin time; aPTT, activated partial thromboplastin time; CK, creatine phosphokinase; CK-MB, creatine kinase Mb; K, potassium; Na, sodium; and CRP, C-reactive protein.

## Data Availability

All of the data is contained within the article.

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
