# Peer review of "Early Blood Glucose Level Post-Admission Correlates with the Outcomes and Oxidative Stress in Neonatal Hypoxic-Ischemic Encephalopathy"

_antioxidants, 2021, doi:10.3390/antiox11010039_

Round 1
Reviewer 1 Report
Early blood glucose level post-admission correlates with the outcomes and oxidative stress in neonatal hypoxic-ischemic encephalopathy
Synopsis: the article is about hypoxic–ischemic encephalopathy in newborns, causes and its treatment. Authors suggest new markers for its early detection.
Critic: There are some flaws in the writing of the article, which nevertheless do not compromise comprehension. Some errors in sentence formulation are noticeable. The bibliography used does not appear particularly current, appearing in some places even outdated. The data shown appear satisfactory for publication requirements.
The P symbol for significance is written sometimes in italic, sometimes not. Can also be found written not in capital in several parts of the article. Authors should standardize.
Text alignment of captions is often incorrect.
The title of paragraphs and/or sections should be corrected, as they are sometimes written in italics, sometimes not.
Sometimes, in the text are found phases or words written in italic without a purpose.
In the results section the numbers are sometimes written in an extended way, sometimes not, sometimes both forms are found; i.e. in lines 188-189 “Among the patients in group 1, two (18.2%) had stage I, seven 7 (63.6%) had stage II, and 2 (18.2%) had stage III HIE”. Same for lines 200-202. As a general rule, numbers one to ten should be written in an extensive way.
Line 74 - the liter symbol must be capitalized.
Line 86 – the sentence: “In men, fasting blood glucose concentration was unchanged in response to acute hypoxia (hours), whereas it increased after 3 days of hypoxia and was restored to normal levels” is not clear.
Line 90 – “A retrospective cohort study of neonates with encephalopathy showed that hypoglycemia was more often associated with watershed hypoxic-ischemic brain injury than basal ganglia hypoxic-ischemic injury, which has been supported in several recent studies [29, 30]”. The reference 30 - “Tam EW, Haeusslein LA, Bonifacio SL, Glass HC, Rogers EE, Jeremy RJ, Barkovich AJ, Ferriero DM: Hypoglycemia is associated with increased risk for brain injury and adverse neurodevelopmental outcome in neonates at risk for encephalopathy. The Journal of pediatrics 2012, 161(1):88-93” can’t be considered so recent.
Lines 98-103 – The concept described in this section “Regarding hyperglycemia, several retrospective studies have reported that hyperglycemia and glucose variability are associated with an increased risk of death and major disability. In hypothermia-treated neonates with HIE, early hypoglycemia or hyperglycemia was noted to cause hearing impairment. In a study of 214 treated neonates with HIE, hyperglycemia was associated with death or unfavorable outcomes at 18 months” is in contradiction with what assessed in lines 81-83 “Hyperglycemia has rarely been reported to present with poor outcomes. HIE is the ischemic and hypoxic change that cause glucose metabolism anomalies”.
Lines 120-122 – it is not clear why some words are written in italic.
Figures 1 and 2 – the space after the equal symbol is not always present.
Figure 1 – the minus symbol must be placed before the number 150.
Figure 1 – there is a space between the / and dL
Lines 134 and 135 – a space is required after the = symbol.
Line 140 – “The”, following a semicolon shouldn’t be written in capital.
Lines 141-144 – the phrase “Brain MRI classified the lesion based on the location: one of the basal ganglion, thalamus, and brain stem (mid-brain, pons, lower brain stem); the other was a lesion located in areas other than the basal ganglion, thalamus, and brain stem” can be written in a better way.
Line 157 – a space in required between the > symbol and the number 1.
Table 1 – a space following = symbol is not always present.
Line 207-209 – the sentence “The remaining cases in group 1 presented 4 (36.4%) with normal brain findings, while 6 (54.5%) exhibited white matter lesions or focal brain lesions that did not involve the thalamus and basal ganglia” can be improved.
Line 214 – why the phrase “Hearing impairment with neonatal HIE were one (8.1%) in group 1, six” is in italic?
Line 221-224 the sentence “In group 1, five (45.5%) had unremarkable and six (54.5%) had abnormal neurodevelopmental outcomes. In group 2, 35 (71.4%) had unremarkable and 14 (28.6%) had abnormal neurodevelopmental outcomes. In group 3, five (35.7%) had unremarkable and nine 9 (64.3%) had abnormal neurodevelopmental outcomes” needs to be written in a better way. Also, the number 9 is written twice, once as a number, once in an extended way.
Lines 231-236 – unclear use of italic for some word. p symbol, on the other hand should be italicized.
Line 240-241 – why the phrase is italicized?
Line 248-250 – the sentence “A significant contribution of this study is its correlation of the first glucose level of neonates with HIE with their clinical staging, imaging findings, hearing outcomes, and neurodevelopmental outcomes at 1 year” should be rewritten in a more correct form.
Line 270 – “First” shouldn’t be written in capital.
Line 285-286 – Why the phrase is in italic?
Author Response
Review 1 Comments and Suggestions for Authors
Early blood glucose level post-admission correlates with the outcomes and oxidative stress in neonatal hypoxic-ischemic encephalopathy
Synopsis: the article is about hypoxic–ischemic encephalopathy in newborns, causes and its treatment. Authors suggest new markers for its early detection.
- Critic:There are some flaws in the writing of the article, which nevertheless do not compromise comprehension. Some errors in sentence formulation are noticeable. The bibliography used does not appear particularly current, appearing in some places even outdated. The data shown appear satisfactory for publication requirements.
Reply: We are grateful for the opportunity to improve our manuscript and we thank the reviewers for their thoughtful and helpful comments and criticisms. We have modified the paper as suggested. Following are our point-by-point responses. We have also highlighted the principal changes.
- The P symbol for significance is written sometimes in italic, sometimes not. Can also be found written not in capital in several parts of the article. Authors should standardize.
Reply: We have used “P” in the Text.
3.Text alignment of captions is often incorrect.
Reply: We have corrected it. The problem is probable due to the technical problem on system of submission.
4.The title of paragraphs and/or sections should be corrected, as they are sometimes written in italics, sometimes not..Sometimes, in the text are found phases or words written in italic without a purpose.
Reply: We have corrected it. The problem is probable due to the technical problem on system of submission.
- In the results section the numbers are sometimes written in an extended way, sometimes not, sometimes both forms are found; i.e. in lines 188-189 “Among the patients in group 1, two (18.2%) had stage I, seven 7 (63.6%) had stage II, and 2 (18.2%) had stage III HIE”. Same for lines 200-202. As a general rule, numbers one to ten should be written in an extensive way.
Reply: We have corrected it as the general rule.
6.Line 74 - the liter symbol must be capitalized.
Reply: We have corrected it.
7.Line 86 – the sentence: “In men, fasting blood glucose concentration was unchanged in response to acute hypoxia (hours), whereas it increased after 3 days of hypoxia and was restored to normal levels” is not clear.
Reply: We have rewritten the sentence, changed to:
“The fasting blood glucose concentration remained unchanged in response to acute hypoxia (hours) [33], but increased after 3 days of hypoxia [39, 40].”
8.Line 90 – “A retrospective cohort study of neonates with encephalopathy showed that hypoglycemia was more often associated with watershed hypoxic-ischemic brain injury than basal ganglia hypoxic-ischemic injury, which has been supported in several recent studies [29, 30]”. The reference 30 - “Tam EW, Haeusslein LA, Bonifacio SL, Glass HC, Rogers EE, Jeremy RJ, Barkovich AJ, Ferriero DM: Hypoglycemia is associated with increased risk for brain injury and adverse neurodevelopmental outcome in neonates at risk for encephalopathy. The Journal of pediatrics 2012, 161(1):88-93” can’t be considered so recent.
Reply: We have deleted “recent”.
9.Lines 98-103 – The concept described in this section “Regarding hyperglycemia, several retrospective studies have reported that hyperglycemia and glucose variability are associated with an increased risk of death and major disability. In hypothermia-treated neonates with HIE, early hypoglycemia or hyperglycemia was noted to cause hearing impairment. In a study of 214 treated neonates with HIE, hyperglycemia was associated with death or unfavorable outcomes at 18 months” is in contradiction with what assessed in lines 81-83 “Hyperglycemia has rarely been reported to present with poor outcomes. HIE is the ischemic and hypoxic change that cause glucose metabolism anomalies”.
Reply: In Introduction, paragraph 3, we have deleted the sentence “Hyperglycemia has rarely been reported to present with poor outcomes. We have rewritten the paragraph, and changed to:
“The normal glucose concentration in the blood of newborn infants is between 2.5 mmol/L (45 mg/dL) and 7.0 mmol/L (126 mg/dL). Most newborns have a blood glucose concentration in the middle of the normal range, approximately 3.5 mmol/L (63 mg/dL) to 5 mmol/L (90 mg/dL) [32]. Hypoxia may cause enhanced or unchanged glucose levels and even decreased or increased concentrations of blood glucose, serum insulin, and plasma glucagon [33-37]. These changes occur since hypoxia was found to stimulate insulin secretion from newborn rats but inhibited in juvenile rats [38]. The fasting blood glucose concentration remained unchanged in response to acute hypoxia (hours) [33], but increased after 3 days of hypoxia [39, 40].”
- Lines 120-122 – it is not clear why some words are written in italic.
Reply: We have corrected it. The problem is probable due to the technical problem on system of submission.
- Figures 1 and 2 – the space after the equal symbol is not always present. Figure 1 – the minus symbol must be placed before the number 150. Figure 1 – there is a space between the / and dL
Reply: We have redone those Figures and added a new Figure (Figure 2) accordingly.
12.Lines 134 and 135 – a space is required after the = symbol.
Reply: We have corrected the problem in all text.
13.Line 140 – “The”, following a semicolon shouldn’t be written in capital.
Reply: We have corrected the problem.
- Lines 141-144 – the phrase “Brain MRI classified the lesion based on the location: one of the basal ganglion, thalamus, and brain stem (mid-brain, pons, lower brain stem); the other was a lesion located in areas other than the basal ganglion, thalamus, and brain stem” can be written in a better way.
Reply: In 2.2. MRI classification, we changed to:
“2.2. MRI classification
The MRI findings were divided into two groups to study the correlation between biomarkers and MRI changes. The first group showed no brain lesions in the parenchyma, while the second group showed brain lesions in the parenchyma. In the second group, brain MRI was classified into two subgroups based on the location of the lesions: one of the basal ganglia, thalamus, or brain stem (midbrain, pons, and lower brain stem) and the other involved areas other than the basal ganglion, thalamus, and brain stem.”
- Line 157 – a space in required between the > symbol and the number 1.
Reply: We have corrected it
- Table 1 – a space following = symbol is not always present.
Reply: We have corrected it,
- Line 207-209 – the sentence “The remaining cases in group 1 presented 4 (36.4%) with normal brain findings, while 6 (54.5%) exhibited white matter lesions or focal brain lesions that did not involve the thalamus and basal ganglia” can be improved.
Reply: We changed those setences to:
“The imaging study in group 1 showed that 1 (9.1%) out of 11 cases exhibited thalamus or basal ganglion lesions. Regarding the remaining 10 cases in group 1, 4 (36.4%) had normal brain findings and 6 (54.5%) exhibited white matter lesions or focal brain lesions that did not involve the thalamus and basal ganglia.”
- Line 214 – why the phrase “Hearing impairment with neonatal HIE were one (8.1%) in group 1, six” is in italic?
Reply: We have corrected it. The problem is probable due to the technical problem on system of submission.
- Line 221-224 the sentence “In group 1, five (45.5%) had unremarkable and six (54.5%) had abnormal neurodevelopmental outcomes. In group 2, 35 (71.4%) had unremarkable and 14 (28.6%) had abnormal neurodevelopmental outcomes. In group 3, five (35.7%) had unremarkable and nine 9 (64.3%) had abnormal neurodevelopmental outcomes” needs to be written in a better way. Also, the number 9 is written twice, once as a number, once in an extended way.
Reply: In 3.5. Correlation of neurodevelopmental outcomes and glucose level. we changed the
“3.5. Correlation of neurodevelopmental outcomes and glucose level
The neurodevelopmental outcome at least 1 year of age was correlated with the first glucose level after admission. In group 1, five (45.5%) had unremarkable and six (54.5%) had abnormal neurodevelopmental outcomes. In group 2, 35 (71.4%) had unremarkable and 14 (28.6%) had abnormal neurodevelopmental outcomes. In group 3, five (35.7%) had unremarkable and nine (64.3%) had abnormal neurodevelopmental outcomes. The neurodevelopmental outcomes in the three groups were significantly different [χ2 (2, n = 74) = 7.11; P = 0.029]. However, the ratio of abnormal neurodevelopmental outcomes at >1 year of age in Groups 1 and 3 was not significant (χ2 [1, n = 25] = 0.24; P = 0.622) (Table 2). In the 72 patients with HIE, glucose levels were significant correlated with neurodevelopmental outcomes [r(72) = 0.331, P = 0.005].”
- Lines 231-236 – unclear use of italic for some word. p symbol, on the other hand should be italicized. Line 240-241 – why the phrase is italicized?
Reply: We have corrected it. The problem is probable due to the technical problem on system of submission.
- Line 248-250 – the sentence “A significant contribution of this study is its correlation of the first glucose level of neonates with HIE with their clinical staging, imaging findings, hearing outcomes, and neurodevelopmental outcomes at 1 year” should be rewritten in a more correct form.
Reply: In Discussion, line 1-3, changed to:
“A significant contribution of this study is the correlation of the first glucose level of neonates with HIE with clinical staging, findings of brain MRI, hearing outcomes, and neurodevelopmental outcomes at 1 year.”
- Line 270 – “First” shouldn’t be written in capital.
Reply: We have corrected it.
- Line 285-286 – Why the phrase is in italic?
Reply: We have corrected it. The problem is probable due to the technical problem on system of submission.

Reviewer 2 Report
Lee and colleagues proposed a research article aimed at evaluating the association existing between glucose levels and the clinical-pathological features of encephalopathy in newborns. For this purpose, the authors retrospectively analyzed the socio-demographic and clinical-pathological features of 74 patients with neonatal HIE as well as the correlation of these features with the baseline glucose levels. Overall, the manuscript is interesting, however, there are some issues that the authors have to address before publication:
1) Please consider providing an Introduction section without subheadings. In addition, the Introduction section should be shortened;
2) Which test was used for correlation analyses? Pearson’s or Spearman’s correlation test should be used;
3) In the main manuscript, please report the data on glucose levels as histograms to a fast and easy visualization of the results obtained;
4) Check the grammar of the first sentence of the Discussion section;
5) In my opinion, the manuscript does not fit well with the topics of antioxidants. The authors are encouraged to stress the pertinence of the manuscript with the aim of the Journal.
Author Response
Reviewer 2 Comments and Suggestions for Authors
Lee and colleagues proposed a research article aimed at evaluating the association existing between glucose levels and the clinical-pathological features of encephalopathy in newborns. For this purpose, the authors retrospectively analyzed the socio-demographic and clinical-pathological features of 74 patients with neonatal HIE as well as the correlation of these features with the baseline glucose levels. Overall, the manuscript is interesting, however, there are some issues that the authors have to address before publication
Reply: We are grateful for the opportunity to improve our manuscript and we thank the reviewers for their thoughtful and helpful comments and criticisms. We have modified the paper as suggested. Following are our point-by-point responses. We have also highlighted the principal changes.
1) Please consider providing an Introduction section without subheadings. In addition, the Introduction section should be shortened
Reply: We have changed it accordingly and shortened the introduction section from 7 paragraphs into 5 paragraphs.
2) Which test was used for correlation analyses? Pearson’s or Spearman’s correlation test should be used;
Reply: We use Persons’s correlation for analysis.
In 2.5. Statistical analysis, changed to:
2.5. Statistical analysis
The independent t-test was performed to compare the means of two independent groups for significant differences between groups, and categorical variables were analyzed using the chi-square test. The Fisher’s exact test was performed when the sample size was small. The odds ratio (OR) was calculated by dividing the odds of the first group by the odds of the second group. Further, the Mann–Whitney U test was performed if the sample distribution was nonparametric, and statistical significance was set at a P-value of < 0.05. For correlation analyses, Pearson’s test was performed to measure the strength of the linear association between the two variables. All statistical tests were performed using SPSS (version 14.0; SPSS Institute, Chicago, IL, USA).
In 3.2. Clinical staging and glucose level, line 8, we added: “Glucose levels were significantly correlated with clinical staging [r (72) = 0.379, P < 0.001].”
In 3.3. Correlation of parenchymal brain lesion and glucose level, line 6-7, we added:
“Glucose levels were significantly correlated with parenchymal brain lesions (r(72) = 0.238, P = 0. 044].”
In line 14-16, we added:
“Of the 28 patients with abnormal lesions on brain MRI, glucose level was significantly correlated with the locations of brain lesions on MRI scans [r(26) = 0.698, P < 0.001].”
In 3.4. Correlation of hearing impairments and glucose level, line 5-6, we added:
“Of the 72 patients with HIE, glucose level was significantly correlated with hearing impairment [r(72) = 0.341, P = 0.003]”.
In 3.5. Correlation of neurodevelopmental outcomes and glucose level, line 9-10, we added “In the 72 patients with HIE, glucose levels were significant correlated with neurodevelopmental outcomes [r(72) = 0.331, P = 0.005].”
In 3.6. The differences of other blood biomarkers in the group 1, group 2 and group 3 patients, line 7-9, we added
“Glucose levels were significantly correlated with LDH [r(64) = -0.401, P < 0.001], SGPT [r(62) = -0.354, P = 0.005], SGPT [r(62) = -0.324, P = 0.010], platelet count [r(67) = 0.208, P = 0.086], and CK [r(67) = -0.235, P = 0.066].”.
3) In the main manuscript, please report the data on glucose levels as histograms to a fast and easy visualization of the results obtained;
Reply: We have added the new Figure (Figure 2) for histogram of glucose level.
4) Check the grammar of the first sentence of the Discussion section;
Reply: We have pleased a professional medical editor who is a native speaker of American English to proofread the text.
In the first sentence of the Discussion section, we changed to “A significant contribution of this study is the correlation of the first glucose level of neonates with HIE with clinical staging, findings of brain MRI, hearing outcomes, and neurodevelopmental outcomes at 1 year.”.
5) In my opinion, the manuscript does not fit well with the topics of antioxidants. The authors are encouraged to stress the pertinence of the manuscript with the aim of the Journal.
Reply: We have added those sentences in Introduction (second paragraph, line 7-19) to fit the topics of antioxidants.
“The pathogenic mechanisms underlying neonatal HIE can be categorized into three phases. The first phase involves primary energy failure due to the hypoxic-ischemic injury, the secondary phase is a consequence of reoxygenation and reperfusion, and the third phase wherein the hypoxic-ischemic injury may worsen and the inflammation may turn into subacute and chronic [21-25] The antioxidant defense system is involved in the pathogenesis of neonatal HIE, particularly in the aforementioned second and third phases [26-28]. During the second phase, the activity of the antioxidant defense system is exhausted due to oxidative stress, leading to further damage, including lipid peroxidation, protein denaturation, enzyme inactivation, and DNA damage [29-31]. Glucose concentration can affect the oxidant-antioxidant balance system in the second and third phases, and impair the antioxidant defense system. Thus, glucose imbalance including hyperglycemia or hypoglycemia is presumed to play an important role in neonatal HIE and also a potential diagnostic and prognostic biomarker.”
In Discussion, paragraph 5, line 13-21, we have written those sentences
“Hyperglycemia in the reoxygenation and reperfusion stage may lead to further brain injury due to consequence of oxidation stress. Hypoglycemia can cause ketogenesis by acting as an alternative cerebral fuel and as antioxidants. This may explain why the hypoglycemia group had better outcomes than the hyperglycemia group in the study. Hyperglycemia caused by insulin resistance can contribute to further brain injury as the consequence of oxidation stress that can be a useful biomarker of poor neurological outcomes and worse neurological consequences [51]. Thus, avoiding hyperglycemia after admission is mandatory in clinical management of neonatal HIE.”
We added the graphical abstract to fit the aim of the Journal.

Reviewer 3 Report
15 December 2021
Review on the manuscript titled “Early blood glucose level post-admission correlates with the outcomes and oxidative stress in neonatal hypoxic-ischemic encephalopathy” by Lee IC et al., submitted to antioxidants
Manuscript ID: antioxidants-1505999
Dear Authors,
The authors studied the relationships between serum blood glucose and neonatal hypoxic-ischemic encephalopathy (HIE) by brain magnetic resonance imaging (MRI), hearing tests, and neurodevelopmental outcomes. The results showed that neonates with abnormal glucose levels lead to brain parenchymal lesion and neurodevelopmental outcomes and that higher glucose levels are associated with worse hearing outcomes. The authors concluded that hyperglycemic neonates have higher chance of developing thalamus, basal ganglia, and brain stem lesions.
Please consider the following:
- A graphical abstract summarizing the manuscript is highly recommended.
- Page 1, Abstract: Background is missing. Please present background, methods, results, and conclusion proportionally.
- Page 1, Keywords: Please list up to ten keywords.
- Pages 1-3, Introduction: Please concisely present birth asphyxia, hypothermic treatment, serum glucose level, and HIE and present details in Discussion.
- Pages 12-14, References: Please cite more references, preferably more than 50 for original articles.
The manuscript contains three figures, three tables and 38 references. The manuscript carries important value presenting serum blood glucose as a potential biomarker for HIE. Thus, I recommend this manuscript for publication after minor revision.
I declare no conflict of interest regarding this manuscript.
Best regards,
Masaru Tanaka, M.D., Ph.D.
Author Response
Reviewer 3
Review on the manuscript titled “Early blood glucose level post-admission correlates with the outcomes and oxidative stress in neonatal hypoxic-ischemic encephalopathy” by Lee IC et al., submitted to antioxidants
Manuscript ID: antioxidants-1505999
Dear Authors,
The authors studied the relationships between serum blood glucose and neonatal hypoxic-ischemic encephalopathy (HIE) by brain magnetic resonance imaging (MRI), hearing tests, and neurodevelopmental outcomes. The results showed that neonates with abnormal glucose levels lead to brain parenchymal lesion and neurodevelopmental outcomes and that higher glucose levels are associated with worse hearing outcomes. The authors concluded that hyperglycemic neonates have higher chance of developing thalamus, basal ganglia, and brain stem lesions.
Reply: We are grateful for the opportunity to improve our manuscript and we thank the reviewers for their thoughtful and helpful comments and criticisms. We have modified the paper as suggested. Following are our point-by-point responses. We have also highlighted the principal changes.
Please consider the following:
- A graphical abstract summarizing the manuscript is highly recommended.
Reply: We have added the graphical abstract.
- Page 1, Abstract: Background is missing. Please present background, methods, results, and conclusion proportionally.
Reply: We have rewritten the abstract according the guideline of the journal and the suggestion.
- Page 1, Keywords: Please list up to ten keywords
Reply: We have added the keywords to ten.
- Pages 1-3, Introduction: Please concisely present birth asphyxia, hypothermic treatment, serum glucose level, and HIE and present details in Discussion.
Reply: We have changed it accordingly and shortened the introduction section to be more concisely. The introduction section has been changed from 7 paragraphs into 5 paragraphs. We also discuss the relevant references in Discussion.
In Discussion, paragraph 5, changed to:
“The findings of the aforementioned studies [28, 29] were compatible with our findings that hypoglycemia and hyperglycemia can increase the risk of poor outcomes in neonatal HIE based on MRI findings. However, in our study, we highlighted the finding that hyperglycemia was associated with a high risk of hearing impairment, which is crucial for childhood neurodevelopment. In hypothermia-treated neonates with HIE for 42 babies, 4 (9.5%) of whom had hearing impairment. The development of hearing loss was associated with abnormal blood glucose levels, low Apgar scores, and evidence of multi-organ dysfunction and increased SGPT and SGPT levels [46], which are compatible with our findings. In addition, we also highlighted that the hyperglycemic patients had more thalamic and basal ganglion injuries than those with hypoglycemia before the first 6 hours. These findings suggest that hyperglycemia can cause selective neuronal necrosis that causes injury to susceptible brain tissue, including the basal ganglia, thalamus, and brain stem. We hypothesized that the mechanism of neonatal HIE is related to glucose and clinical staging (Figure 4). Hyperglycemia in the reoxygenation and reperfusion stage may lead to further brain injury due to consequence of oxidation stress. Hypoglycemia can cause ketogenesis by acting as an alternative cerebral fuel and as antioxidants. This may explain why the hypoglycemia group had better outcomes than the hyperglycemia group in the study. Hyperglycemia caused by insulin resistance can contribute to further brain injury as the consequence of oxidation stress that can be a useful biomarker of poor neurological outcomes and worse neurological consequences [51]. Thus, avoiding hyperglycemia after admission is mandatory in clinical management of neonatal HIE.”
- Pages 12-14, References: Please cite more references, preferably more than 50 for original articles.
Reply: We have added the references. The references number now is more than 50.
The manuscript contains three figures, three tables and 38 references. The manuscript carries important value presenting serum blood glucose as a potential biomarker for HIE. Thus, I recommend this manuscript for publication after minor revision.
Reply: Thanks for the feedback.
